# Envisaging Antifungal Potential of Histatin 5: A Physiological Salivary Peptide

**DOI:** 10.3390/jof7121070

**Published:** 2021-12-12

**Authors:** Pratibha Sharma, Mehak Chaudhary, Garima Khanna, Praveen Rishi, Indu Pal Kaur

**Affiliations:** 1University Institute of Pharmaceutical Sciences, Panjab University, Chandigarh 160014, India; pratibhapulastya@gmail.com (P.S.); mehakchaudhary1999@gmail.com (M.C.); garimakhanna10@gmail.com (G.K.); 2Department of Microbiology, Panjab University, Chandigarh 160014, India

**Keywords:** fungi, histatin 5, candidiasis, *Candida*, periodontal, oral cavity, structure activity relationship

## Abstract

Fungi are reported to cause a range of superficial to invasive human infections. These often result in high morbidity and at times mortality. Conventional antifungal agents though effective invariably exhibit drug interactions, treatment-related toxicity, and fail to elicit significant effect, thus indicating a need to look for suitable alternatives. Fungi thrive in humid, nutrient-enriched areas. Such an environment is well-supported by the oral cavity. Despite this, there is a relatively low incidence of severe oral and periodontal fungal infections, attributed to the presence of antimicrobial peptides hosted by saliva, viz. histatin 5 (Hstn 5). It displays fungicidal activity against a variety of fungi including *Candida albicans, Candida glabrata, Candida krusei, Cryptococcus neoformans*, and unicellular yeast-like *Saccharomyces cerevisiae*. *Candida albicans* alone accounts for about 70% of all global fungal infections including periodontal disease. This review intends to discuss the scope of Hstn 5 as a novel recourse for the control of fungal infections.

## 1. Introduction

Fungal infections impact the quality of life in the inflicted patients [1]. Further, these infections display a multifaceted spectrum because of their diverse extent and severity [2,3]. *Candida albicans* is documented as the prime agent for mucosal and systemic infections. It is responsible for about 70% of all fungal infections [4] including periodontal infections.

The mainstay of treatment lies with azoles, polyenes, and echinocandins. Azoles such as fluconazole and itraconazole inhibit ergosterol biosynthesis in the fungal cell membrane [5]. Polyenes viz. amphotericin B bind to the membrane ergosterol molecule and cause pores in the fungal cell membrane [6]. Echinocandins inhibit fungal cell wall biosynthesis. Azoles possess moderate tolerance but are associated with substantial drug interactions. Polyenes suffer from low oral bioavailability and renal impairment. Echinocandins elicit adequate fungicidal action but exhibit a narrow spectrum [7]. Howsoever, even in the presence of these antifungal chemotherapeutics, the mortality rate is as high as 40% [8]. Drug development of novel antifungal agents is challenging as fungi show high similarity with the host cells [9]. All these factors support the ever-increasing demand for more advanced antifungal agents. The oral cavity is the most exposed avenue for microbial entry to the body. Structures inside it are continuously bathed in saliva. Fungi thrive in humid, nutrient-enriched areas of the oral cavity. Despite this, there is relatively low incidence of severe oral and periodontal fungal infections. These observations indicate a significant antifungal potential of saliva [10].

Saliva is an extracellular fluid synthesized and secreted by salivary glands inside the oral cavity. It is viscoelastic in nature and vital for managing oral health and homeostasis. It is composed of water (99.5%), ions, proteins, and enzymes [11]. Antimicrobial agents such as lysozymes, hydrogen peroxide, lactoferrin, and histatins compose the protein fraction of saliva [12,13]. Xerostomia, the condition of the oral cavity under reduced saliva production, is often marked by oral candidiasis and periodontal disease. Latter is attributed to depletion of saliva and hence its low antimicrobial effects.

Histatins (Hstns) are histidine-rich, low molecular weight, cationic, antimicrobial peptides present in human saliva and the oral cavity. Hstns exhibit a broad spectrum of antifungal activities and reportedly play a significant role in controlling periodontal and oral fungal infections [14,15]. Manytimes, patients with periodontal and oral infections show an elevated level of Hstns which may be a physiological response of the body to counter these persistent infections. Till date 26 Hstns have been documented in human saliva. Out of which Hstn 1, Hstn 3 and Hstn 5 account for more than 80% of the total Hstn concentration. Hstn 1 and Hstn 3 are full-length products translated by *His1* and *His2* located on chromosome 4, respectively. Hstn 1 undergoes proteolysis to produce Hstn 2, while proteolysis of Hstn 3 leads to the production of all other subtypes [16].

Out of all Hstns, Hstn 5 shows strong antifungal potential against *C. albicans*, the most prevalent fungi causing periodontal disease, skin, and other superficial infections [12,17,18]. Hstn 5 activates the release of fungal mitochondrial ATP which binds to purinergic receptors causing fungal cell death [19].

Hstn 5 is a naturally occurring peptide in humans, with no reported cross-reactivity in any human cells or tissues. Overall Hstn 5 showcases significant potential as a novel and highly effective antifungal agent [20]. A significant number of patents exist on Hstn 5 and its applications (Table 1).

A plethora of bioactivity displayed by Hstn 5 stimulates the compilation of this review which describes the structure and source of this salivary peptide and its potential as a novel antifungal therapeutic recourse, its purported mechanism of action, and plausible applications in diverse fungal infections including oral and periodontal infections.

## 2. Structure-Function Studies on Hstn 5

Peptides and their structural modification to produce products with high bioavailability and commercial applicability has been a subject of continuous interest for the scientific community.

The structure of Hstn 5 is Asp-Ser-His-Ala-Lys-Arg-His-His- Gly-Tyr-Lys-Arg-Lys-Phe-His-Glu-Lys-His-His-Ser-His-Arg-Gly-Tyr. It is represented as DSHAKRHHGYKRKFHEKHHSHRGY. Hstn 5 though comprising of only 24 amino acids yet shows the highest candidacidal activity among all Hstns [26]. The structural flexibility of Hstn 5 allows it to attain different conformations [27]. In the oral cavity, it exists in a linear state but acquires alpha-helical conformation in the hydrophobic environment of the Candida cell membrane [28]. Hstn 5 forms a structure with almost all the positive charges on one side and a non-polar face on the other side of the tightly coiled helical structure. This amphipathic nature of the Hstn 5 is critical for its activity. Helical conformation and C-terminal sequence are the functional domains of Hstn 5. However, the helical conformation is not directly correlated with antifungal activity. A Hstn 5 variant 3P has reduced potential for helical conformation but shows antifungal potential comparable to Hstn 5. It is developed by substituting three residues of Hstn 5 with proline [29]. The fungicidal activity of Hstn 5 lies in the C-terminal sequence (11–24), a 14-residue fragment. This fragment is known as dh-5 and the sequence comprises of seven histidine, four lysine, and three arginine amino acids [28]. These cationic residues are fundamental for its antifungal activity. P113 is the smallest variant of Hstn 5, comprising of 12 amino acids that exhibits anticandidal activity. A two-fold increase in the activity is reported after amidation on its C terminus sequence. Due to its smaller size, P113 has a reduced propensity for helical conformation as compared to parent Hstn 5. Moreover, the fungicidal activity is not compromised with certain structural modifications in P113. Substitution of lysine or arginine at position 3 or 9 with glutamine leads to a reduction in the fungicidal activity of the peptide. Similarly, the substitution of glutamine with either lysine or arginine at positions 2 or 10 resulted in the loss of fungicidal activity [30]. Similar observations have been reported in variants M21 and M71 developed by substituting the lysine at position 13 with threonine and glutamic acid, respectively. Both variants’ manifest reduced fungicidal properties. This corroborates the significance of lysine and arginine in fungicidal activity [31]. Studies on structural modification of Hstn 5 are summarized in Table 2.

## 3. Mechanism of Action

Hstns, being cationic amphipathic molecules with alpha-helical conformation target the anionic microbial surface [32]. Hstn 5 is documented to target the intracellular components of fungi [33]. The fungicidal action of Hstn 5 has been reported to be a multi-step process (Figure 1) which is discussed in the subsequent sections.

### 3.1. Binding and Uptake of Hstn 5

Three mechanisms are proposed for the uptake of Hstn 5 which potentially work in tandem. In addition to the uptake after a direct interaction with the plasma membrane, transport-mediated and receptor-mediated endocytosis are the other two proposed uptake pathways [34]. The concentration of the Hstn 5 is the main factor influencing these pathways.

Lower extracellular concentration directs the operation of receptor-mediated endocytosis thus internalizing Hstn 5in the vacuoles. Higher physiological concentrations lead to Hstn 5 accumulation in the cytoplasm through transport mediated uptake. However, at intermediate concentrations, the accumulation of this peptide inside the cell is not site-specific [32,35]. The complex structure of the fungal cell wall makes the utilization of transporter proteins and adenosine triphosphate (ATP) for Hstn 5 transit across it. Heat shock protein, cytosolic Ssa 1p/2p is identified for Hstn binding [36,37]. These binding proteins are present in the cell envelope of *C. albicans* and are involved in traversing Hstn 5 across the plasma membrane. Binding occurs in an energy-independent fashion and is required for localized retention before energy-mediated uptake. The fungal polyamine transporters Dur3 and Dur31 eventually actively transport Hstn 5 into the cells [38].

### 3.2. Efflux of Intracellular Contents

Hstn 5 causes exudation of intracellular potassium ions (K^+^) [39] leading to an ionic imbalance in the fungal cell, eventually causing cell death [40]. Trk 1 transporters are documented to participate in this mechanism. Cells lacking the Trk 1 transporters are quite resistant to the fungicidal action of Hstn 5. Simultaneously, there is a loss of viability due to volume dysregulation in the cells. Ionic imbalance is found to be the plausible exposition for the loss of volume from the fungal cells [41].

### 3.3. ATP and Mitochondrion Mediated Cell Death

Release of intracellular ATP after incorporation of Hstn 5 is also considered to be a cause of cell death in *C. albicans*. This released ATP mediates cell killing by binding to the purinergic receptors of the cell membrane, P2X [42]. Thus, intracellular ATP reduction and its efflux ultimately lead to fungal cell death [43]. The mitochondrion is a widely studied intracellular target that is involved in cell killing. It is documented that active metabolism is obligate for Hstn 5 mediated cell killing in *C. albicans*. Cellular respiration blocked by sodium azide [44] or other classical mitochondrial inhibitors rendered the cells resistant to Hstn 5 mediated killing. Similarly, the energy deficit cells are reported to be resistant to this peptide-mediated killing due to the perturbations induced in their lipid bilayer as a consequence of lowered energy [45]. Studies establish the relationship between the release of reactive oxygen species (ROS) and Hstn 5 mediated cell death. The impairment of respiratory activity leads to the production of ROS and is believed to be the reason for the self-inflicted cell suicide mechanism [19]. Thus, it can be concluded that mitochondrion plays a significant role in the internalization of Hstn 5 by maintaining the membrane potential [46].

### 3.4. MAPK Signalling Induced Cell Death

The mitogen-activated protein kinases (MAPK) signaling pathwayis stimulated by many antifungal drugs [47]. Hstn 5 induced oxidative and osmotic stresses are the plausible reasons to engender stress MAPK Hog1 in Hstn 5 treated cells. This furnishes the evidence that oxidative stress-induced cell death cannot be negated completely. Hstn 5 treated *C. albicans* cells showed phosphorylation of Hog1. This induces the production of glycerol resulting in osmotic stress [48]. These observations corroborated the intended role of induction of osmotic stress as a mediator for the candidacidal activity of Hstn 5. Recent studies have shown that the cell wall integrity, MAPK Cek1, and Ssa2 binding protein also play a significant role in the fungicidal action of Hstn 5 [34]. Cek1 is responsible for controlling the production of Hstn 5 binding counterparts in the fungal cell wall [49]. Phosphorylated Cek1 and N-acetylglucosamine are the only carbon sources making the cells susceptible to Hstn 5 [50]. Consequently, the increased localized retention of Hstn 5 leads to fungal cell death. However, there can be a condition where *C. albicans* degrades Hstn 5 by employing aspartyl proteases, thus hampering its fungicidal activity [51].

## 4. Antifungal Potential of Hstn 5

A goal of modern therapeutics is to develop non-invasive antifungal agents to prevent fungal growth. In this regard, Hstn 5, has attracted considerable interest in many therapeutic areas.

### 4.1. Oral Candidiasis

Oral candidiasis is a fungal infection affecting the oral mucosa caused by the overgrowth of *C. albicans* in the oral cavity. It is a common fungal disease observed in immunocompromised patients [52]. However, the emergence of resistant fungal pathogens necessitates the development of novel therapeutics. The growth of oral candidiasis is reported to be regulated by saliva and its components. Among all salivary peptides, Hstn 5 and its derivatives are documented to be the major peptides that kill *C. albicans* [26]. Hstn 5 is evidenced to clear lesions and the associated tissue inflammation in infections of the mouth [53]. P113, a derivative of Hstn 5 is documented to be effective against the fluconazole-resistant strains [54]. Likewise, dentures incubated with Hstns resulted in a notable reduction of *C. albicans* contamination [55]. Thus, in the light of these observations, Hstn 5 could be a potent agent for improving oral health in immunocompromised patients without any adverse effects [56].

### 4.2. Periodontal Disease

Periodontal disease constitutes a series of inflammatory conditions related to the tooth and its supporting tissues. Approximately 10% of the global population is affected by severe periodontitis [57,58]. In the initial stage, there is a plaque formation that affects the gingiva causing redness, inflammation, and bleeding. In later stages, the formation of periodontal pockets takes place which ultimately leads to loss of tooth attachment [59]. *C. albicans* is reported to be one of the prime causative organisms for periodontal disease. The carriage of this pathogen is directly related to the severity of the condition [60]. *C. albicans* in synergy with bacteria invade the dental tissues resulting in the proliferation and exudation of enzymes that eventually cause tissue degradation [61]. Biofilm formation is considered to be a crucial virulence factor of Candida species [62]. These biofilms are extremely resistant to antimicrobial agents and to the host immune responses rendering them one of the major obstacles to overcome in the treatment of periodontal disease. Currently, systemic antibiotic therapy and removal of bacteria mechanically are the mainstay treatment employed [63,64]. However, the increased incidence of resistance requires the development of novel therapeutics. As Hstn 5 is a part of the innate host defense system and has potent antimicrobial and antifungal properties, it can thus be employed as an intervention for periodontal disease [65]. Hstn 5 and its derivatives reduce gingivitis, bleeding, and plaque formation. P113 is reported as the most effective among all the studied agents [66]. Hstn 5 act on the exponential growth phase of infecting organisms. Furthermore, no adverse events and side-effects were observed indicating the safety of these agents [67]. All these studies support that Hstn 5 can be a potential auxiliary agent to treat periodontitis.

### 4.3. Vulvovaginal Candidiasis

Vulvovaginal candidiasis is an inflammatory disease, affecting the female genital tract. A retrospective study, including over 950,000 women who followed gynecological practices showed that approximately 70–75% of the women population experience vaginal candidiasis at least once in their lifetime, and up to 50% of them suffer from recurrent candidiasis. Vulvovaginal candidiasis is caused by the uncontrolled growth of Candida yeast species [68]. Although various species of genus Candida are involved in the pathogenesis of this disease, *C. albicans* is involved in nearly 76–89% of all the infection cases being reported [69]. Use of oral contraceptives, diabetes mellitus, pregnancy, and long-term broad-spectrum antibiotic treatment is recognized as the predisposing factors which encourages the growth of yeast [70]. Signs and symptoms associated with this disease are vaginal discharge, pruritus, discomfort during urination, pain during sexual intercourse, and erythematous vulva [71]. Fluconazole is usually employed as the mainstay treatment but the widespread increase in the resistant strain results in the relapse of this disease [68]. A study was conducted on 52 women suffering from this disease to check the susceptibility of yeast against the available antifungal drugs. Surprisingly, 49 (94%) of the isolates were identified as the fluconazole-resistant *C. albicans* strains which indicate the possible role of drug-resistant strains in recurrent episodes of disease [72]. In addition, fluconazole only lengthens the asymptomatic period but does not completely eradicate the causative agent from the affected area [73]. Hence the failure of therapy employing conventional antifungal encourages the need to identify or develop novel agents. Antimicrobial peptides constitute an innate host defense system against pathogens [74]. The antimicrobial property of Hstn 5 against oral candidiasis and dentistry is remarkable [75]. Similarly, a wide range of antimicrobial peptides are present in the healthy female genital tract so that it seems appropriate to evaluate the potential of Hstn 5 in the treatment of vulvovaginal candidiasis [76]. Hstn 5 retains its antifungal activity at pH < 3.8 thus it will be a good candidate for vaginal delivery where the pH is <4.5. Antifungal potential of Hstn 5 has been evaluated against clotrimazole in a murine model of vulvovaginal candidiasis. Hstn 5 is reported to show a substantial decline of fungal burden and reduce the vaginal discharge in the murine model. All these observations encourage its use in the topical treatment of vulvovaginal candidiasis [69].

## 5. Conclusions

Peptides are naturally occurring macromolecules and closely mimic the intrinsic signaling entities which make them highly contemporary therapeutics. Further, a lot of work has been undertaken to design novel peptides and suitable delivery systems and penetration enhancers to ensure half-life augmentation and better permeation of these agents. All this has collectively broadened the scope of the application of peptides.

Hstn 5, a cationic histidine-rich salivary peptide of human and higher primates has shown a promising potential to reduce mycological morbidity among immunocompromised patients. Different mechanisms are reported to support its antifungal activity. Further, the physiological nature of Hstn 5 ensures the absence of toxicity. While there is substantial evidence on the use of Hstn 5, however, a lot is desired on developing a suitable delivery system for the same to maintain its stability without compromising efficacy. The future thus foresees a bright perspective for the inclusion of novel drug delivery system which can facilitate the promise of this peptide.

## Figures and Tables

**Figure 1 jof-07-01070-f001:**
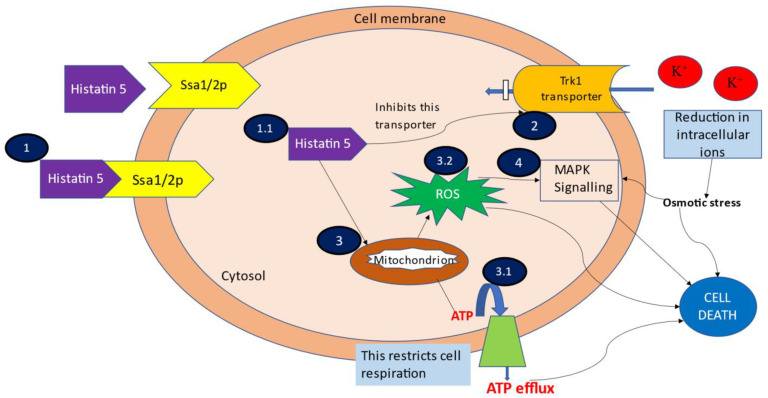
Mode of action of Hstn 5 via different mechanisms (**1**) Binding and uptake of Hstn 5; (**1.1**) Targeting of intracellular components; (**2**) Efflux of intracellular contents; (**3**) ATP and mitochondrion mediated cell death; (**3.1**) Exudation of ATP; (**3.2**) Generation of ROS; (**4**) MAPK signaling induced cell death.

**Table 1 jof-07-01070-t001:** Patents on Hstn 5 and its derivatives.

Patent Number	Description	Highlights	Reference
US 2014/0065119A1	The invention focuses on the use of cyclic analogues of Hstn 5 for the treatment of wounds. Cyclization improves stability and cellular uptake of Hstn 5. Cyclable amino acids can be incorporated to induce cyclization in Hstn 5 and its derivatives.	Therapeutically effective doses range from 0.01 mg to 100 mg per kg of body weight. A suitable absorbent hydrogel can be developed for topical application. Hstn 5 along with other therapeutic agents can be used for wound healing.	[21]
WO 2016/060916 A1	The invention focuses on the utilization of Hstns as therapeutic agents for ocular surface diseases such as dry eye. Hstn 5 being a modulator of inflammatory cytokines can be incorporated in anti-inflammatory formulations along with other therapeutics.	The preferred weight to weight ratio of Hstn5:cHstn 1 was 1:1, 6:1, 1:10, 1:15. Hstn 5 and Hstn 1 are combined in ranges from 1 µg to 10 mg/mL. Both Hstns were mixed with 0.1% to 1% glycerin to form sterile eye drops. Hstn 5 along with rapamycin can be administered to treat dry eye in patients suffering from autoimmune diseases such as Sjogren’s syndrome.	[22]
US 7781531 B2	Dentures conventionally made from poly (methyl methacrylate) lead to denture-induced stomatitis in the user due to adhesion of *C. albicans*. This invention focuses on the incorporation of Hstn 5 with phosphate-containing co-polymers in the dentures. Phosphate anion facilitates the adhesion of cationic Hstn molecules overdenture for restraining the induced complications.	Adsorption of Hstn 5 increases with an increase in the negative charge on the polymer.	[23]
WO 2009/005798 A2	The invention describes Hstn 5 derivative based mouth rinse formulation for improved antifungal activity.	Amidation at the carboxyl terminus of the Hstn 5 derivative resulted in a two-fold increase in antimicrobial activity.	[24]
US 2010/0202983 A1	The invention describes the utilization of carrier agents for the delivery of Hstn and its derivatives for the treatment of periodontal disease.	Carrier agents and Hstns are covalently coupled to form a complex. Formed complex ensures sustained release of Hstn with better penetration and retention.	[25]

**Table 2 jof-07-01070-t002:** The structural variants of Hstn 5 and their corresponding antifungal activity.

Variant	Sequence	Structural Modifications Compared to Hstn 5	Reported Activity compared to Hstn 5	Reference
P113	AKRHHGYKRKFH–NH_2_	12 amino acid sequence amidated on C terminus Reduced propensity to make an alpha helix	Two-fold increase in fungicidal activity after amidation LD_50_ = 2.3 ± 0.65 µg/mL	[30]
M21	DSHAKRHHGYKRTFHEKHHSHRGY	Lysine at 13 position substituted with threonine	Reduced fungicidal activity	[31]
M71	DSHAKRHHGYKREFHEKHHSHRGY	Lysine at 13 position substituted with glutamic acid	Reduced fungicidal activity	[31]

## Data Availability

Not applicable.

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
