# Peer review of "Envisaging Antifungal Potential of Histatin 5: A Physiological Salivary Peptide"

_jof, 2021, doi:10.3390/jof7121070_

Round 1

Reviewer 1 Report

The authors present a review paper on the important role of salivary antimicrobial peptide Histatin-5. Overall the paper is not well written, is difficult to follow with incoherent sections, and features significant grammatical issues. Important basic science research citations are missing from the literature and also form the patent literature. Inaccuracies in microbiology terminology and factual inaccuracies are evident throughout the manuscript. Unclear from which source Fig.1 was adapted from. The paper does not add anything substantial as compared to with is currently available in the literature. 

Author Response

Paper is revised to eliminate the grammatical issues.

Relevant citations are placed after every point.

Patent table number 1 is revised.

Microbiology terminology is revised and so the complete manuscript is revised accordingly.

Figure 1 is not adapted from any source. It is designed by us only.

Reviewer 2 Report

The authors of the manuscript entitled “Envisaging Antifungal Potential of Histatin 5: A Physiological Salivary Peptide” aimed to provide a simple review of histatin 5 as a potential antifungal drug. Globally, the work is interesting however, due to the poor use of English, it is hard to read. Also, there are important issues that must be corrected and clarify.

Major issues

Line 31 – “Dermatophytes namely Candida species and Malassezia species…” This is a huge and serious mistake: neither Candida spp nor Malassezia spp are Dermatophytes!!! Dermatophytes are filamentous fungi (Trichophyton, Epidermophyton and Microsporum) that are responsible for cutaneous mycosis. Please correct the sentence

Line 32 – Candida spp is responsible for cutaneous and opportunistic mycoses, not superficial mycoses!

Line 37 – The azoles act in the cell membrane not in the cell wall. Please clarify.

Line 38-39- It depends on the site of infection. Echinocandins are the first line therapeutical option regarding invasive candidiasis. Please clarify.

Line 46-49 – Traditional antifungal agents? Please clarify

Line 48 – The sentence does not make sense. Please revise.

Line 51 – Please revise “…as well as…”.

Line 81 – Candida spp is not responsible for superficial mycoses.

Line 101, 140, 168, 230, 308, …please correct the section numbers.

Line 205-207 – This sentence does not make sense. Please correct.

Line 220 – “Diagrammatic”? Please delete

Line 248 – “opportunistic agent”? Please correct.

Author Response

Paper is revised to eliminate the grammatical and other issues.

Line 31 – Revised

Line 32 – Revised

Line 37 – Revised

Line 38-39- Revised

Line 46-49 – Revised

Line 48 – Revised

Line 51 – Revised

Line 81 – Revised

Line 101, 140, 168, 230, 308, Revised

Line 205-207 – Revised

Line 220 – Revised

Line 248 – Revised

Reviewer 3 Report

In this review article submission the authors focus on the antifungal potential of histatin 5, a human peptide that can kill Candida directly. The need for novel antifungals is high, and therefore it is important to understand the therapeutic potential of the histatin family of peptides. In its current form this review contains  English language usage and grammar errors that significantly distract from the content of the manuscript. There are also sections that can be expanded on in order to provide a better understanding of histatins in human health and disease as outlined in the following points:

  • There are countless English grammatical errors and usage issues. Including, but not limited to: incorrect noun/verb tense usage, use of colloquialisms, etc. In the least this is quite distracting, and at times causes confusion with the point the authors are trying to make (line 102). 
  • It is not clear how Candida prefers growth in the human oral cavity (37C) when the optimum growth range is 25 to 30. This needs to better explained for readers who may not understand. 
  • It is not clear why HIV/AIDS patients would have salivary defects (lines 66-70).  Explanation is needed. 
  • References appear to be missing in several places. Including, but not limited to, line 91.
  • Table I appears extraneous. The information about the patents, while relevant, does not warrant the expanded use of a table. Also, not all the abbreviations used in the table are explained. 
  • Some information in the introduction is contradictory. The authors rightly explain that the incidence of fungal infections are on the rise, but that oral candidiasis is rare except in cases of xerostomia or other salivary defects. This is misleading and to a certain degree contradicts section 1.1. The authors need to clarify. 
  • The authors should include in the introduction more detail about the other histatin family members. 
  • In 1. Mechanism of action the author explain the action of histatin 5 is a multi-step process which is described in subsequent sections. But, it is not clear enough that this is what is described in the subsections that follow. The sections do not seem to be connected in anyway (ie as steps).
  • The mechanisms are outlined in Figure 1 as 7 mechanisms, but this is not clear in the text. Perhaps number the mechanisms in this way manuscript not just the figure. 

Author Response

Paper is revised to eliminate the grammatical issues.

Line 102: Revised

Passage regarding candida growth in oral cavity, on page 1 is revised

Line 66-70: Revised

Line 91: Revised

Table 1: Revised

Section 1 and 4.1 is revised

Introduction on histatin family is included

Hstn 5 can act in multiple ways (four described in text) including some sub-mechanisms of section 3.1 and 3.3. Hence the figure number 1 shows 7 pathways.

Round 2

Reviewer 1 Report

The changes and modifications are not sufficient to warrant publication. This paper adds little new information and is not different from current reviews on the subject.

Author Response

The manuscript extensively revised again.

This peptide has immense potential for development as a novel antifungal agent specially to address recurring and resistant infections. This review is an effort to compile all reports on Hstn 5 and sensitize the scientific community on its potential.

No such review article compiling scope and existing patent application on Hstn 5 is there.

Through this review we also want to highlight that most patents and studies report on use of Hstn 5 and its structural modification but there could be a tremendous opportunity to develop formulation/ products for patient by taking it from the lab to the shelf.

Reviewer 2 Report

The authors have addressed reasonably my recommendations. However extensive editing of English is mandatory.

Author Response

The manuscript is extensively revised to address all grammatical errors.

Reviewer 3 Report

In this resubmission the authors have addressed many of my concerns, but there still remains grammatical and English language usage issues. These problems need to be resolved prior to acceptance of the manuscript for publication. In many cases the language issues lead to confusion and change the authors' intent. While not an exhaustive list, many are listed below:

Line 45 "This serves". What serves? Needs to be explained. Also furnish may not be the best choice of words. 

Line 63 "while hstn3 led to the production of..." How did hstn 3 lead to production? Not sure what the authors' intent is. 

Lines 78-84 This is an awkward way to phrase this, and meaning is lost because of this. 

Line 86 "It may be" 

Throughout: Candida should be capitalized. Within a sentence, candidiasis should not. Same for oral. 

Line 113 "These studies are summarized in table 2", should be the last sentence of the paragraph. 

Line 136. Remove "the" from "the Candida albicans". This mistake needs to be corrected throughout the manuscript.

Line 148 "to be the cause of death". Don't you mean "a" cause of death. 

Line 152 Active should not be capitalized. 

Line 157 "Also there are many evidences". Incorrect phrasing. 

Line 16 "environmental induced". I am not sure what the authors intent is using this phrase. 

Line 196 "The intervention". What intervention?  The antecedent is not clear. Fix this grammatical error  throughout the manuscript. 

Line 201 phrases such as "was evidenced" are extraneous. Please use active voice throughout the manuscript. There are numerous issues like this. 

Line 210. "pathogen" Structure of sentence including use of pathogen is not understood. 

Line 214. antecedent unclear

Line 219 remove "to be"

Lines 225-226 phrasing should be: Hstn5 and its derivatives reduce gingivitis. Again, this occurs throughout the manuscript. 

Line 230 "these all studies". Incorrect phrasing. 

Line 233 use affecting the female genital tract

Line 241 are recognized

Line 248 "its" Not sure what this means. antifungal resistance? The role of antifungal resistance in recurrent episodes of disease? 

Line 257-258. Strange phrasing. Meaning is lost.

Author Response

The manuscript extensively revised to address all grammatical errors.

Line 45 Revised

Line 63 Revised

Lines 78-84 Revised

Line 86 Revised

Candida is capitalized throughout the manuscript. Candidiasis and  oral are also revised according to the manuscript need.

Line 113 Revised

Line 136. Revised

Line 148 Revised

Line 152 Revised

Line 157 Revised

Line 16 Revised

Line 196 Revised

Line 201 Revised

Line 210. Revised

Line 214. Revised

Line 219 Revised

Lines 225-226 Revised

Line 230 Revised

Line 233 Revised

Line 241 Revised

Line 248 Revised

Line 257-258. Revised